# Nanoparticulate carbon black in cigarette smoke induces DNA cleavage and Th17-mediated emphysema

Ran You[1,2,3], Wen Lu[1,2,3], Ming Shan[1], Jacob M Berlin[4,5], Errol LG Samuel[6], Daniela C Marcano[6], Zhengzong Sun[6], William KA Sikkema[6], Xiaoyi Yuan[1], Lizhen Song[1], Amanda Y Hendrix[1], James M Tour[6]*, David B Corry[1,2,3,7]*, Farrah Kheradmand[1,2,3,7]*

[1]Department of Medicine, Baylor College of Medicine, Houston, United States; [2]Department of Pathology and Immunology, Baylor College of Medicine, Houston, United States; [3]Biology of Inflammation Center, Baylor College of Medicine, Houston, United States; [4]Department of Molecular Medicine, Beckman Research Institute, City of Hope National Medical Center, Duarte, United States; [5]Irell & Manella Graduate School of Biological Sciences, City of Hope National Medical Center, Duarte, United States; [6]Department of Chemistry, Rice University, Houston, United States; [7]Michael E. DeBakey VA Center, US Department of Veterans Affairs, Houston, United States

**Abstract** Chronic inhalation of cigarette smoke is the major cause of sterile inflammation and pulmonary emphysema. The effect of carbon black (CB), a universal constituent of smoke derived from the incomplete combustion of organic material, in smokers and non-smokers is less known. In this study, we show that insoluble nanoparticulate carbon black (nCB) accumulates in human myeloid dendritic cells (mDCs) from emphysematous lung and in CD11c$^+$ lung antigen presenting cells (APC) of mice exposed to smoke. Likewise, nCB intranasal administration induced emphysema in mouse lungs. Delivered by smoking or intranasally, nCB persisted indefinitely in mouse lung, activated lung APCs, and promoted T helper 17 cell differentiation through double-stranded DNA break (DSB) and ASC-mediated inflammasome assembly in phagocytes. Increasing the polarity or size of CB mitigated many adverse effects. Thus, nCB causes sterile inflammation, DSB, and emphysema and explains adverse health outcomes seen in smokers while implicating the dangers of nCB exposure in non-smokers.

*For correspondence: tour@rice. edu (JMT); dcorry@bcm.edu (DBC); farrahk@bcm.edu (FK)

**Competing interests:** The authors declare that no competing interests exist.

**Reviewing editor**: Feng Shao, National Institute of Biological Sciences, China

## Introduction

Tobacco smoking is linked to a long and growing (*Barnes, 2014*; *Carter et al., 2015*) list of fatal illnesses (e.g., emphysema, cancer, and stroke) and is the major preventable cause of human death. Despite public awareness of the harmful effects of smoking, in many large developing countries the prevalence of smoking is growing (*Eriksen et al., 2014*). Compounding this risk is particulate air pollution due to the combustion of organic materials including biomass fuels, slash and burn agriculture, and coal (*Furlaneto et al., 1969*; *Arif et al., 1993*; *Dadvand et al., 2014*). While our understanding of the immune basis of smoke-induced sterile inflammation has increased, the molecular mechanism underlying emphysema and its persistence despite smoking cessation remains unclear (*Cosio et al., 2009*; *Kheradmand et al., 2012*). Even less is known regarding the health-related inhalation effects of atmospheric and workplace airborne carbon particulates.

Innate immune cells such as alveolar macrophages and neutrophils are recruited to the lungs in response to cigarette smoke (*Salvi, 2014*). Several human studies and pre-clinical models of smoke-induced

**eLife digest** Smoking for many years damages the lungs and leads to a disease called emphysema that makes it difficult to breathe and is often deadly. There are thousands of chemicals in cigarette smoke and many of them have been linked to the development of lung cancer, although it has been difficult to pinpoint those that are responsible for smoking-related emphysema. Moreover, cigarette smoke also contains large numbers of small particles and relatively little is known about the role played by these particles in smoking-related disease.

One of the hallmarks of long-term smoking is a blackening of the lung tissue that persists even if someone stops smoking. Previously, little was known about the composition of the substance that causes this blackening, or its significance in the development of emphysema. Now, by studying lung tissue taken from smokers with emphysema, You et al. have shown that this black substance is made of nano-sized particles of a material called carbon black (which is also known as elemental carbon). These nanoparticles are produced by the incomplete combustion of the cigarettes. You et al. also confirmed that nanoparticles of carbon black can cause emphysema in mice.

Closer examination of the lung damage caused by the nanoparticles revealed that they trigger breakages in DNA, which leads to inflammation of the lung. And because the nanoparticles cannot be cleared, they are released into the lung when cells die, which perpetuates lung inflammation and damage.

You et al. then went on to show that nanoparticles of carbon black can be modified in a way that allows them to be cleared from the lungs. Such modifications could potentially protect people who are exposed to carbon black nanoparticles in the environment or in workplaces where carbon black is used, such as factories that produce automobile tires and other rubber products.

emphysema have also confirmed that lymphocytes (T and B cells) and lung antigen presenting cells (APCs) play pathogenic roles in chronic lung inflammation in smokers (*Shan et al., 2009*, *2014*; *Churg et al., 2012b*). Prior work has shown that increased concentrations of pro-inflammatory cytokines such as IL-6, IL-1β, and IL-17A are among the most important hallmarks of smoke-mediated lung inflammation (*Shan et al., 2009*, *2014*; *Chang et al., 2014*). We and others have previously shown that IL-17A plays a critical role in smoke-induced emphysema in humans and in mouse models of disease (*Shan et al., 2012*; *Kurimoto et al., 2013*; *Zhang et al., 2013*; *Chang et al., 2014*). Further, adoptive transfer of lineage-negative CD11c+ myeloid dendritic cells (mDCs) isolated from the lungs of smoke-exposed mice to naive mice recapitulates smoke-induced lung disease, indicating a direct causal relationship between mDC and emphysema (*Shan et al., 2012*). Despite these advances, the mechanism by which smoke induces the inflammatory mDC phenotype remains completely unknown.

Tobacco smoke contains many noxious chemicals (e.g., carbon monoxide, sulfur, nitrogen dioxide, nitric oxide, and methane), aromatics (e.g., benzene, toluene, and xylene) and chlorinated (e.g., methyl chloride, chloroethene, and chloroform) volatile organic compounds, as well as particulate matter (*Wang et al., 2012*; *Perfetti and Rodgman, 2013*; *Salvi, 2014*). One or more of these agents is thought to underlie the carcinogenic potential of smoke, involving at least eight different cancers; accordingly, the role of volatile carcinogens found in smoke has been studied extensively (*Pope et al., 2011*; *Carter et al., 2015*). Far less is known about the pathogenic effects of particulate matter that is suspended in smoke and which includes nanoparticulate carbon, metal oxides, and inorganic salts. Histopathological analysis of the lungs of heavy smokers invariably reveals dark-staining anthracotic pigment often attributed to poorly soluble material found in tobacco smoke (*Mitchev et al., 2002*). Anthracotic pigment is also found in the lymph nodes of smokers (*Churg et al., 2005*), but its chemical composition and potential contribution to smoking-related diseases have not been explored.

In this study, we show that the anthracotic material found in the lung of human smokers is nanoparticulate carbon black (nCB), a hydrophobic material that accumulates specifically in highly activated CD1a+ lung APCs. Raman spectroscopy, hyperspectral imaging, and high-resolution transmission electron microscopy (HR-TEM) were used to confirm this observation and show that nCB further accumulates in the lung and APC of mice exposed to smoke. Moreover, we show that nCB

given by inhalation to mice in amounts that are comparable to human exposures is alone sufficient to cause sterile inflammation and induce robust T helper 17 cell (Th17) responses and emphysema, implicating the potential health risks of airborne nCB that contaminates a wide range of workplace and domestic environments (*IARC Working Group on the Evaluation of Carcinogenic Risks to Humans, 2010*).

## Results

### Detection of nCB in lungs of smokers

In contrast to the white or pink appearance of normal lungs, the lungs of heavy smokers are typically dark brown or black (*Churg et al., 2005*). During the routine preparation of lung phagocytic cells including CD1a$^+$ mDCs and APCs, we observed the same anthracotic pigmentation in lung cells from smokers (*Figure 1A,B*), whereas the same cells isolated from non-smokers lacked the pigment. We previously showed that activated mDCs from smokers and smoke-exposed mice promote Th1 and Th17 responses (*Shan et al., 2009*, *2012*). To determine the nature of the anthracotic pigment from mDCs, we first performed HR-TEM of the residual black material after complete proteolytic digestion of human emphysematous lung and observed the aggregates that were composed of 20–50 nm spheroids (*Figure 1C*). To further extend these observations, we examined CD1a$^+$ mDCs from human emphysematous lung tissue using Raman spectroscopy and hyperspectral mapping which showed the signature for pure nCB and not inorganic salts or metal oxides (*Figure 1D–H*). Thus, nCB is extensively deposited as solid material in the lungs of smokers and specifically accumulates in lung APC.

Next, using our established mouse model of cigarette smoke-induced emphysema (*Shan et al., 2012*, *2014*), we exposed mice to 4 cigarettes daily or air for 4 months to examine whether nCB accumulates in lung APCs. When compared to air-exposed mice, we confirmed that isolated CD11c$^+$ APCs and cells present in bronchoalveolar (BAL) fluid (macrophages >90%) (*Shan et al., 2012*) contain particulate matter with the Raman spectral signature of nCB within each cell (*Figure 1I*). These findings indicate that both humans and mice chronically exposed to cigarette smoke accumulate nCB within phagocytic lung APCs.

### nCB induces emphysema in mice

We have previously recapitulated smoke-induced lung sterile inflammation and emphysema by adoptively transferring lineage-negative CD11c$^+$ mDCs isolated from the lungs of smoke-exposed mice to naive mice, which revealed the direct, causal role of mDCs in emphysema (*Shan et al., 2012*). As we found that these mDCs contained nCB, we sought to determine if nCB was alone sufficient to induce emphysema. We first determined that the commercial nCB does not desorb polycyclic aromatic hydrocarbons (PAHs), as determined by Soxhlet extraction followed by gas chromatography mass spectroscopy (GCMS) (*Harwood and Moody, 1989*). Mice were then exposed twice weekly for 6 weeks to hydrocarbon free, hydrophobic, neutral surface charged nCB (average particle size 15 nm), to achieve a total lung dose of ~1% of wet lung weight (mg/g), which approximates human lung nCB burdens (*Figure 2—figure supplement 1*).

4 weeks after the last intranasal instillation of nCB, harvested lungs were extensively anthracotic, similar in appearance to lungs of long-term smokers (*Figure 2A*). Physiologically, the nCB challenge-induced enlargement of the alveolar spaces (*Figure 2B*) concomitant with significant increases in total lung volume quantified by micro-CT imaging and unbiased lung morphometry measurement (mean linear intercept; MLI) (*Figure 2C,D*), both hallmarks of emphysema. Mice exposed to nCB showed significantly increased numbers of macrophages, neutrophils, and lymphocytes in BAL fluid as compared to vehicle (PBS)-challenged control animals (*Figure 2E*). Consistently, increased lung inflammation was accompanied by higher concentrations of inflammatory cytokines and chemokines (*Figure 2—figure supplement 2*) as well as elastolytic matrix metalloproteinases (MMPs) 9 and 12 (*Figure 2F*), all of which are characteristic features of cigarette smoke-induced emphysema in human patients and animal models of this disease (*Shan et al., 2009*; *Churg et al., 2012a*). Similarly, both lung parenchymal CD11c$^+$ mDCs and BAL fluid macrophages showed an accumulation of nCB as detected by hyperspectral imaging (*Figure 2G–I*).

Gross and microscopic examination of the lungs at 7 and 18 months following the last nCB exposure showed persistence of nCB, lung inflammation, and hyperinflation

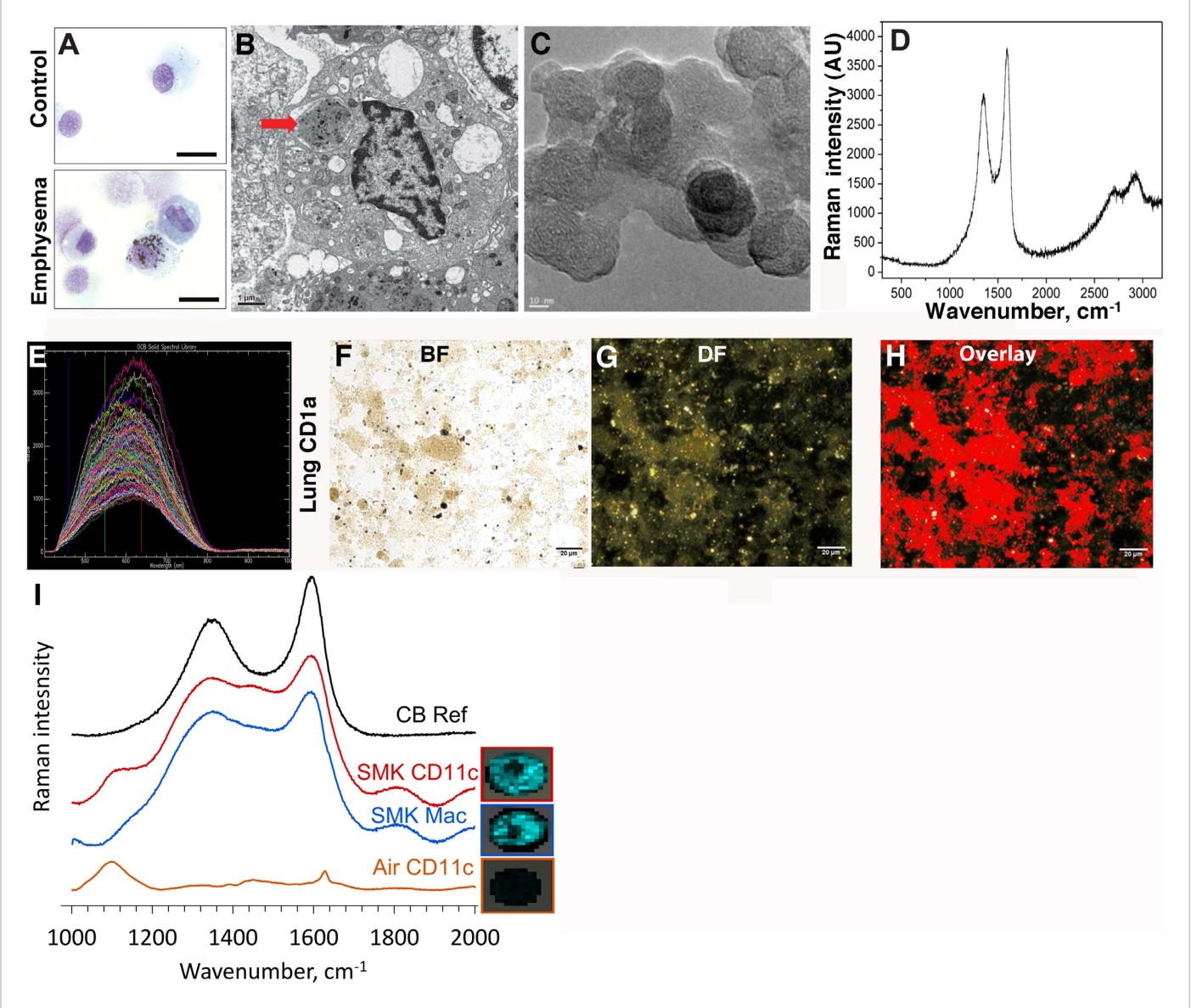

**Figure 1**. Carbon black (CB) deposition in the lungs of patients with emphysema. (**A**) Representative images of lung CD1a+ cells from a smoker with emphysema and a control subject. Scale bar: 10 µm. (**B**) Lung CD1a+ cells from a patient with emphysema, detected by transmission electron microscopy (TEM). Arrow indicates black substance in the vesicles. Scale bar: 1 µm. (**C**) Structure of the residual black material from digested human emphysema lung tissue, detected by high-resolution transmission electronic microscopy (HRTEM). Scale bar: 10 nm. (**D**) Raman spectrum yielded by the black material in the cells. The bifid spectral peaks between 1000 and 2000 cm$^{-1}$ are the typical Raman signature for CB. Representative hyperspectral image of lung CD1a+ cells from a patient with emphysema (**E–H**): a reference sample of nanoparticulate carbon black (nCB) was used to generate a signature spectral library (**E**) using CytoViva Hyperspectral Imaging System. Each colored spectra represents the spectral profile of a distinct area of the nCB sample, which were used in combination to map nCB present in cells. (**F**) Bright field (BF), (**G**) dark field (DF), and (**H**) overlay CB signature spectrum of lung CD1a+ cells. Positive signals were pseudo-colored red to aid visualization. Scale bar: 20 µm. (**I**) Raman spectrum yielded in lung CD11c+ and macrophages isolated from lungs of mice exposed to smoke for 4 months; CB reference (CB Ref) signal indicates solid CB sample. SMK: 4 months of cigarette smoke. Inset images for cell type correspond to Raman spectra indicating the subcellular localization of CB. The brightness of each 2 µm × 2 µm pixel, representing one spectrum, indicates the height of the graphitic band of CB at 1600 cm$^{-1}$ compared to the background, such that brighter pixels indicate more CB.

(*Figure 2—figure supplement 3–5*). Thus, nCB—as an insoluble byproduct of tobacco combustion as shown above and also delivered through a non-smoking model—accumulates in lung and airway APCs, is poorly cleared from the lungs, and is alone sufficient to cause lung inflammation and emphysema in mice.

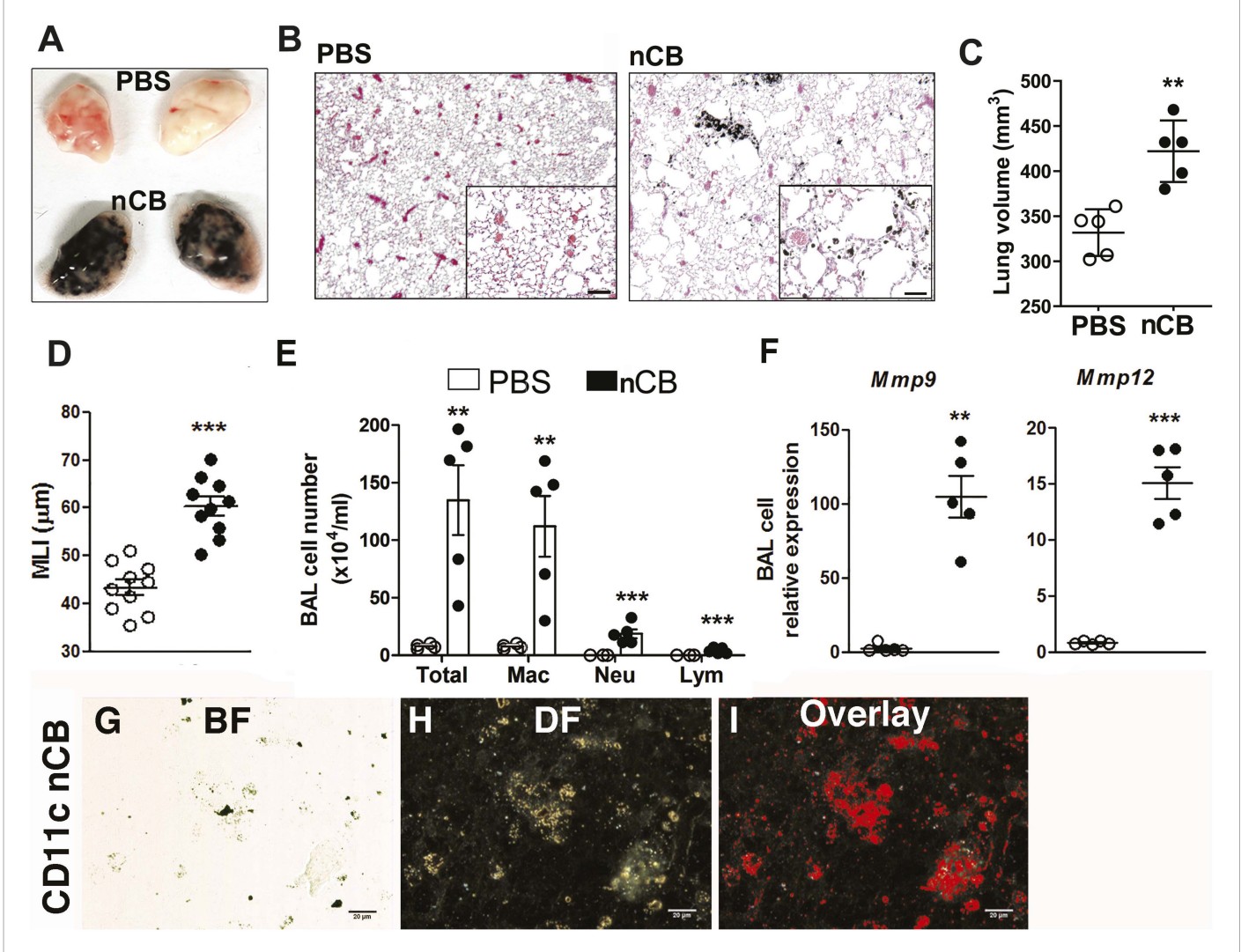

**Figure 2**. Carbon black-induced emphysema mouse model. (**A**) Representative image of fresh lungs harvested from mice exposed to vehicle (PBS) or nanoparticulate carbon black (nCB) as described in *Figure 2—figure supplement 1*. (**B**) Representative Hematoxylin and eosin (H&E) staining of formalin-fixed lung sections. Scale bar: 100 μm. (**C**) Micro-CT quantification of lung volume. (**D**) MLI measurement was done on the same groups of mice. (**E**) Total and differential cell count in bronchoalveolar (BAL) fluid: macrophages (Mac), neutrophils (Neu), and lymphocytes (Lym). Quantitative PCR of *Mmp9* and *Mmp12* (**F**) gene expression in BAL cells isolated from PBS- or CB-challenged mice. Representative lung CD11c+ cells isolated from mice challenged with nCB under bright field (BF) (**G**), dark field (**H**), and overlap images (pseudo-red area) (**I**) signifying nCB signature spectrum. Scale bar: 20 μm. Data are mean ± SEM and representative of three independent experiments; ***p < 0.001, **p < 0.01 as determined by the Student's *t*-test; n = 5 per group.

The following figure supplements are available for figure 2:

**Figure supplement 1**. Schematic representation of nCB-induced lung inflammation and emphysema protocol.

**Figure supplement 2**. nCB induces pro-inflammatory cytokines and chemokines in the lung.

**Figure supplement 3**. nCB persists in the lungs 18 months after the last challenge.

**Figure supplement 4**. nCB-induced emphysema persists in the lungs.

**Figure supplement 5**. nCB-induced immune cell infiltration persists in the lungs.

## nCB activates APCs to secrete pro-Th17 cytokines and inhibits regulatory T cell differentiation in vitro

We previously determined that cigarette smoke induces lung APC activation in human patients and mice, which then induces Th17 cell differentiation in naive T cells (*Shan et al., 2009*). To determine whether nCB specifically induces Th17 responses in vivo, we first examined lung mDCs from nCB intranasal-challenged mice. Lung CD11c$^+$CD11b$^{hi}$ mDCs were significantly increased in the lungs of nCB-challenged mice when compared with controls (*Figure 3A,B*). nCB also selectively induced lung Th17 but not Th1 responses relative to control animals (*Figure 3C,D* and *Figure 3—figure supplement 1*). Lung CD11c$^+$ APCs isolated from nCB-challenged mice secreted significantly more of the Th17 cell growth factors IL-6 and IL-1β, along with other pro-inflammatory cytokines and chemokines, but not IL-12 or IL-4 (IL-4 was undetectable in both PBS and nCB groups), which promote Th1 and Th2 cell differentiation, respectively (*Figure 3—figure supplement 2*). To determine if lung APCs from nCB-challenged mice induce specific T cell differentiation programs in vitro, we co-cultured naive splenic CD4$^+$ T cells with CD11c$^+$ cells isolated from lungs of nCB- or PBS-challenged mice. Lung APCs from nCB-challenged mice induced significantly more IL-17A, but neither IFN-γ nor IL-4 production, when compared to controls (*Figure 3—figure supplement 3*). Lung Th17 responses persisted for at least 7 months following the last nCB challenge (*Figure 3—figure supplement 4*). Further, *Il-17a$^{-/-}$* mice were resistant to nCB challenge as assessed by their attenuated increases in lung volume, lung immune cell infiltration, and the reduced destruction of alveoli (*Figure 3E–H*) when compared to identically treated WT mice. Thus, in vivo nCB selectively induces chronic lung Th17 responses, which are crucial for CB-induced emphysema in mice.

We next explored whether nCB plays a direct role (i.e., independent of APCs) on T helper cell differentiation. To address this question, we polarized T cells toward Th1, Th17, and regulatory (Treg) phenotypes in the presence or absence of nCB in vitro. We found that nCB did not affect Th1 or Th17 cell differentiation directly (*Figure 3—figure supplement 5A*). However, nCB treatment significantly inhibited Treg differentiation (*Figure 3—figure supplement 5A,B*). These findings indicate that nCB promotes sterile inflammation by inducing Th17 differentiation indirectly through APCs and directly by inhibiting Treg differentiation.

## Hydrophobicity of nCB correlates with pathogenicity

The previous findings demonstrated that when deposited in the lungs, nCB activates mDCs and induces durable Th17-dependent inflammation and emphysema in mice. To determine the mechanism of nCB-mediated lung pathology, we next investigated whether its physicochemical properties could account for its immunostimulatory function. Whether manufactured or found in the lungs of smokers with emphysema, nCB is very hydrophobic and completely insoluble in aqueous media. Conjugating polyethylene glycol to nCB (PEG-nCB) renders the material hydrophilic and miscible with aqueous solutions (*Hwang et al., 2014*). Mice challenged with intranasal PEG-nCB using the same protocol (*Figure 2—figure supplement 1*) failed to develop emphysema as assessed by quantitative CT-based lung volume measurements, MLI and microscopic evaluation of the lungs (*Figure 4A,B,C*). Further, we detected less anthracotic pigment in the lung parenchyma, suggesting that in contrast to hydrophobic nCB, PEG-nCB could be cleared from the lungs (*Figure 4C*). Microscopic inspection of isolated BAL fluid cells from PEG-nCB-challenged mice showed intact phagocytic cells compared to that of hydrophobic nCB, suggesting that the latter may induce less cytotoxic effects on phagocytic cells (*Figure 4—figure supplement 1*). In support of this, we found that the release of lactate dehydrogenase (LDH), an indicator of cytotoxicity, was enhanced in macrophage-like RAW 264.7 cells exposed to nCB as compared to PEG-nCB (*Figure 4—figure supplement 2*).

Consistent with the failure to induce emphysema and cell death, exposure to PEG-nCB also resulted in attenuated recruitment of macrophages, neutrophils, and lymphocytes to the lung when compared with hydrophobic nCB (*Figure 4D*). This reduction in inflammation was accompanied by reduced expression of *Mmp9* and *Mmp12* transcripts in BAL fluid cells as compared with nCB-challenged mice (*Figure 4E,F*). The markedly reduced inflammatory nature of PEG-nCB was further underscored by the reduced concentrations of pro-inflammatory cytokines and chemokines detected from freshly collected lung homogenates of PEG-nCB-challenged mice (*Figure 4—figure supplement 3*), including decreased IL-6 and IL-1β levels (*Figure 4G,H*). Critically, PEG-nCB failed to

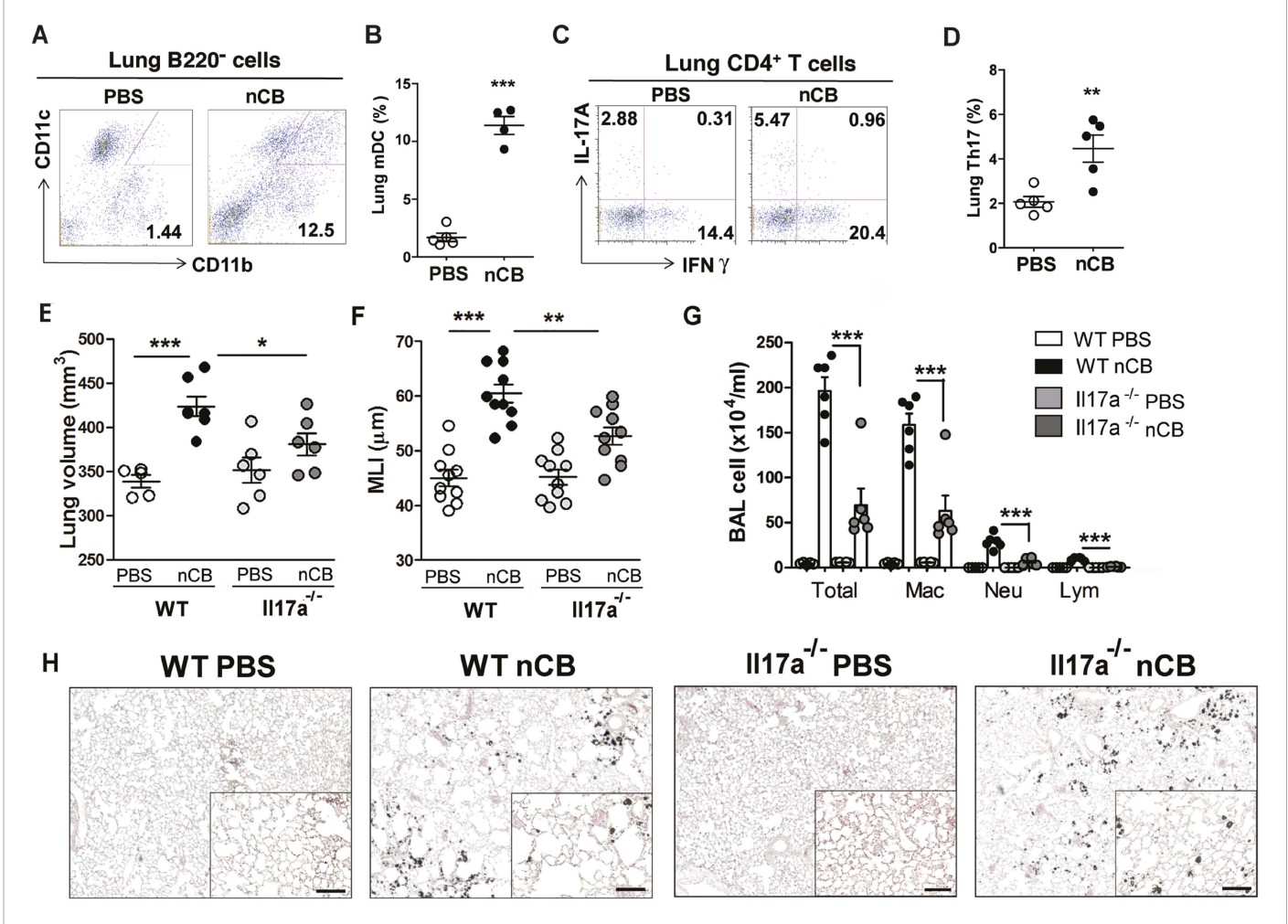

**Figure 3**. nCB promotes Th17 responses. Representative staining (**A**) and cumulative analysis (**B**) of the percentage of CD11c+CD11b$^{high}$ cells in lung B220− cell subset. Representative intracellular staining (**C**) and cumulative analysis (**D**) of IL-17A+ cells expressing lung CD4+ T cell (Th17) subset. (**E**) Micro-CT quantification of lung volume in WT and Il-17a$^{−/−}$ mice. (**F**) Lung MLI was determined in the same group of mice. (**G**) BAL fluid analysis of the indicated groups of mice showing the total cells including macrophages (Mac), neutrophils (Neu), and lymphocytes (Lym). ***p < 0.001, **p < 0.01, *p < 0.05 as determined by the one-way ANOVA and Bonferroni's multiple comparison test. N = 4 to 6 per group. Data are mean ± SEM. (**H**) Representative H&E staining of formalin-fixed, 5-μm lung sections in indicated groups of mice. Scale bar: 100 μm.

The following figure supplements are available for figure 3:

**Figure supplement 1**. nCB did not induce Th1 responses.

**Figure supplement 2**. Lung APCs of nCB-challenged mice secrete Th17 cell-specific pro-inflammatory cytokines and chemokines.

**Figure supplement 3**. Lung APCs of nCB-challenged mice-induced Th17 responses.

**Figure supplement 4**. nCB-induced Th17 responses persist in the lungs.

**Figure supplement 5**. Direct effect of nCB on T helper cell differentiation in vitro.

induce lung Th17 cells when compared to nCB-exposed animals (*Figure 4I,J*). Thus, the pro-inflammatory potential of nCB is intimately tied to its hydrophobic surface and ability to induce cytotoxicity of phagocytic cells.

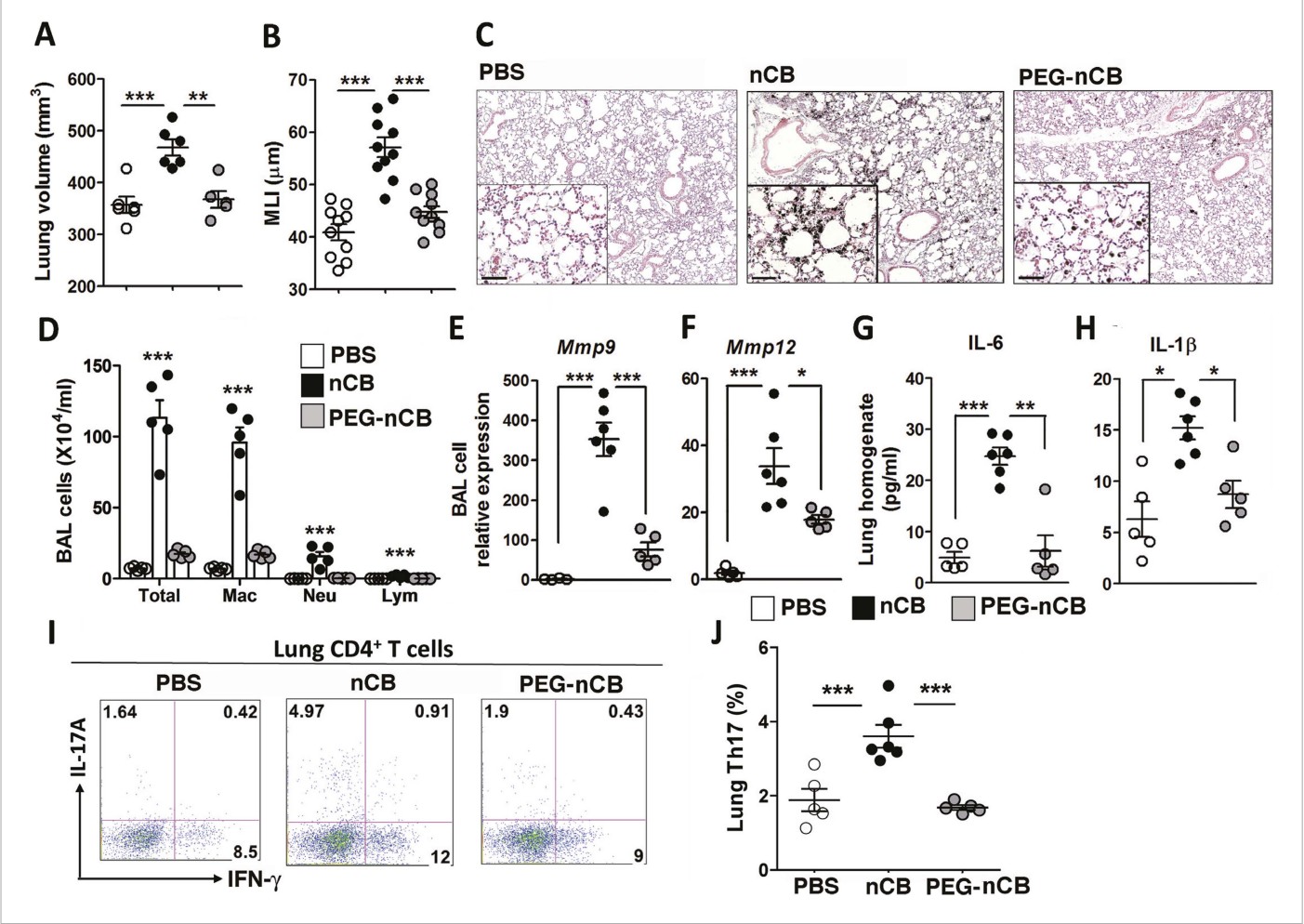

**Figure 4**. Hydrophobicity of nCB is important for its pathogenesis. Micro-CT quantification of lung volume (**A**) and MLI measurement of lung morphometry (**B**) in vehicle (PBS), nCB, and PEG-nCB treated mice. (**C**) Representative H&E staining of lung sections Scale bar: 100 μm. (**D**) Total and differential cell count in bronchoalveolar (BAL) fluid; macrophages (Mac), neutrophils (Neu), and lymphocytes (Lym). Quantitative PCR of *Mmp9* (**E**) and *Mmp12* (**F**) gene expression in BAL cells isolated from the above group of mice. Lung homogenate collected from indicated groups of mice were measured for IL-6 (**G**) and IL-1β (**H**) by ELISA. Representative intracellular staining (**I**) or cumulative analysis (**J**) of Th17 cells in the lungs. ***p < 0.001, **p < 0.01, *p < 0.05 as determined by the one-way ANOVA and Bonferroni's multiple comparison test. n = 4 to 6 per group, and data are mean ± SEM and representative of two independent studies.

The following figure supplements are available for figure 4:

**Figure supplement 1**. nCB-induced cell damage compared with PEG-nCB.

**Figure supplement 2**. nCB-induced cell death compared with PEG-nCB.

**Figure supplement 3**. nCB-induced strong lung inflammation compared with PEG-nCB.

## nCB-mediated induction of DNA damage and Erk signaling activates APCs

We conducted additional studies to determine how nCB activates APCs to secrete pro-inflammatory cytokines (e.g., IL-6 and IL-1β) and chemokines. In response to nCB, but not PEG-nCB, reverse phase protein array (RPPA) identified the activation of several DNA damage (e.g., PARP, p-Chk2, p-ATM) and MAPK/Erk (p-ERK, p-MEK1/2)-response proteins (*Figure 5A* and *Figure 5—figure supplement 1*). Consistent with these data, we found that nCB, but not PEG-nCB, induced DNA double strand breaks

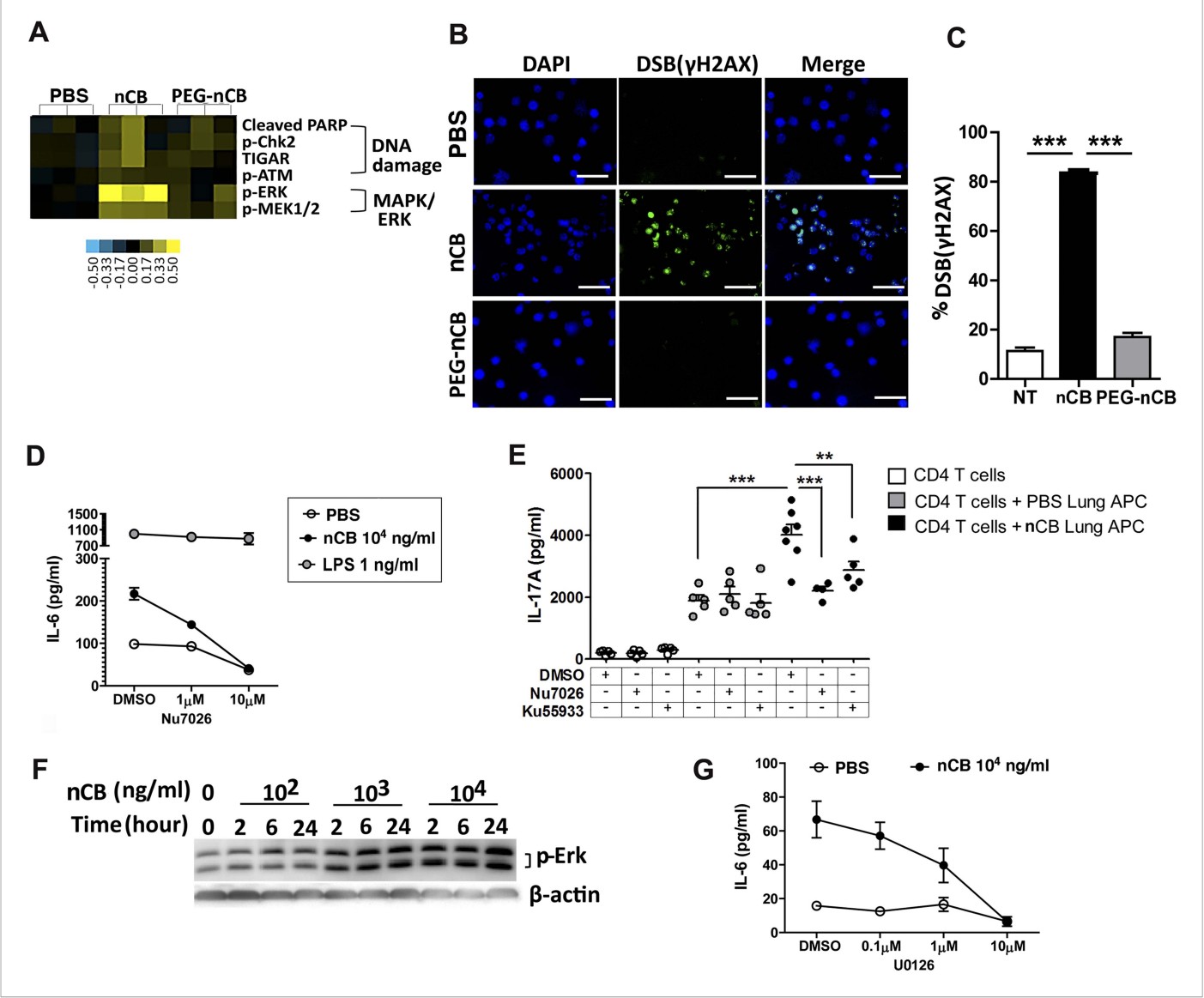

**Figure 5**. nCB activates APCs by the induction of DNA damage and Erk signaling. (**A**) Heat map (reverse phase protein array) of protein expression and phosphorylation level in RAW 264.7 cells stimulated with vehicle (PBS), nCB ($10^5$ ng/ml), and PEG-CB ($10^5$ ng/ml). p: phosphorylated. Blue is relatively low (−0.5) and yellow high (0.5) based on log2 ratio of the value for expression level. (**B**) RAW 264.7 cells under indicated conditions immunostained for nuclear DNA (DAPI, blue) and anti-γH2AX (green) to detect double strand break (DSB). Scale bar: 50 μm. (**C**) Quantitative summary of panel **B** indicating the percentage γH2AX positive RAW cells in indicated groups. (**D**) IL-6 concentration detected by ELISA after 48 hr in the supernatant of MDDC treated with CB or LPS in the presence of increasing dose of Nu7026 or vehicle (DMSO). (**E**) IL-17A concentration detected by ELISA after 72 hr co-culture of splenic CD4 T cells and lung CD11c$^+$ cell isolated from the mice after challenged with PBS or nCB and anti-CD3 (1 μg/ml) in the presence of Nu7026 (100 nM), Ku55933 (100 nM), or vehicle control (DMSO). (**F**) Western blot of protein extracted from BMDC treated with different concentration of nCB targeting phosphorylated-Erk. Data are representative of two independent experiments. (**G**) IL-6 concentration detected by ELISA in the supernatant of MDDC treated with nCB in the presence of increasing dose of U0126 (MEK1/2 inhibitor) for 48 hr. n = 4 to 7 per group and data are mean ± SEM and representative of two independent experiments (**C**, **D**, **E**, **G**). ***p < 0.001, **p < 0.01 as determined by the one-way ANOVA and Bonferroni's multiple comparison test.

The following figure supplements are available for figure 5:

**Figure supplement 1**. Heat map depicting molecules whose expression and phosphorylation level differed when RAW 264.7 cells were treated with nCB compared with PBS or PEG-nCB treated groups detected by reverse phase protein array.

**Figure supplement 2**. Larger nCB size correlates with weak induction of DNA double strand breaks (DSB).

*Figure 5. continued on next page*

Figure 5. Continued

**Figure supplement 3**. ATM is required for nCB-induced inflammatory factor upregulation in RAW cells.

**Figure supplement 4**. Inhibition of DNA damage does not affect Th1 or Th2 responses.

(DSB) as determined by phosphorylation of Histone 2AX (H2AX) on serine 129 ($\gamma$H2AX) (*Figure 5B,C*). Further, the induction of DSB was inversely dependent on the size of nCB as we observed progressively fewer DSB with increasing nCB size (*Figure 5—figure supplement 2*). We next examined whether CB-induced DSB could account for the pro-inflammatory responses seen in APC. Human monocyte-derived dendritic cells (MDDCs) treated with Nu7026, an inhibitor of the DNA-dependent protein kinase catalytic subunit (*Wilmore et al., 2004*; *Zhou et al., 2014*), exhibited reduced IL-6 production in a dose-dependent manner in response to nCB but not LPS (*Figure 5D*). Moreover, in nCB-exposed RAW 264.7 cells, transfection of a specific siRNA against ataxia telangiectasia mutated (ATM)—a serine–threonine kinase that coordinates repair of double-stranded DNA breaks (*Guo et al., 2010*)—significantly reduced expression of IL-6 and TNFα, two inflammatory cytokines that are induced through ATM (*Figure 5—figure supplement 3*).

To further examine whether induction of Th17 responses is dependent on nCB-mediated DNA damage, CD11c+ lung mDCs isolated from nCB-challenged mice were co-cultured with splenic CD4 T cells in the presence of either Nu7026 or Ku55933, an inhibitor of ATM (*Li and Yang, 2010*), for 3 days. As expected, mDCs isolated from nCB-challenged mice promoted Th17 cell differentiation, which was significantly reduced in response to Nu7026 or Ku55933 (*Figure 5E*) while Th1 and Th2 cell differentiation remained unchanged (*Figure 5—figure supplement 4*). Together, these findings suggest that nCB-mediated DNA damage is required for the induction of pro-inflammatory cytokines in mDCs and Th17 cell differentiation. Moreover, nCB exposure in a dose- and time-dependent way increased phosphorylation of Erk (*Figure 5F*), and similar inhibition of MEK1/2 with U0126, an inhibitor of MAP kinases (*Newton et al., 2000*), reduced IL-6 production in response to nCB exposure (*Figure 5G*). Together, these findings indicate that hydrophobic nCB activates DNA damage responses and induces MAPK/Erk signaling coincident with the induction of Th17 responses.

## ASC-mediated assembly of the inflammasome complex is required for nCB-induced Th17 responses and emphysema

The inflammasome detects danger signals released in response to cell injury and sterile inflammation and the adaptor protein ASC (apoptosis-associated speck-like protein containing CARD) was shown to be required for inflammasome-dependent caspase-1–mediated conversion of pro-IL-1β to mature IL-1β (*Kono et al., 2012*). In response to nCB exposure, lung CD11c+ mDCs increased IL-6 and IL-1β expression and RAW 264.7 cells released more LDH, consistent with the concept that nCB induces both sterile inflammation and necrotic cell death. To determine if ASC is also required for nCB-induced Th17 responses and emphysema, *Pycard*−/− mice were challenged intranasally with nCB. When compared to WT mice treated identically, *Pycard*−/− mice showed attenuated emphysema (*Figure 6A–C*) and reduced macrophage, neutrophil, lymphocyte, and mDC infiltration into the lungs (*Figure 6D,E*). Consistently, lung mDCs of *Pycard*−/− mice produced less IL-6 and IL-1β and poorly activated splenic T cells to differentiate into Th17 cells when compared with WT mDC (*Figure 6F–H*). Freshly collected lung homogenates from *Pycard*−/− mice challenged with nCB also showed reduced inflammatory chemokine production compared with WT mice (*Figure 6—figure supplement 1*). Thus, the earliest immunological events induced by nCB include ASC activation and inflammasome assembly, which are in turn required for nCB-mediated Th17 responses and emphysema.

## Discussion

Evidence from experimental systems and human translational studies strongly support a role for chronic inflammation—and Th17 cells in particular—in the initiation and progression of emphysema in smokers (*Shan et al., 2012*; *Eppert et al., 2013*; *Kurimoto et al., 2013*). A characteristic feature of the anthracotic pigment of smokers' lungs is that such discoloration persists even long after smoking has ceased (*Churg et al., 2005*). In this study, we addressed the role of insoluble anthracotic pigment

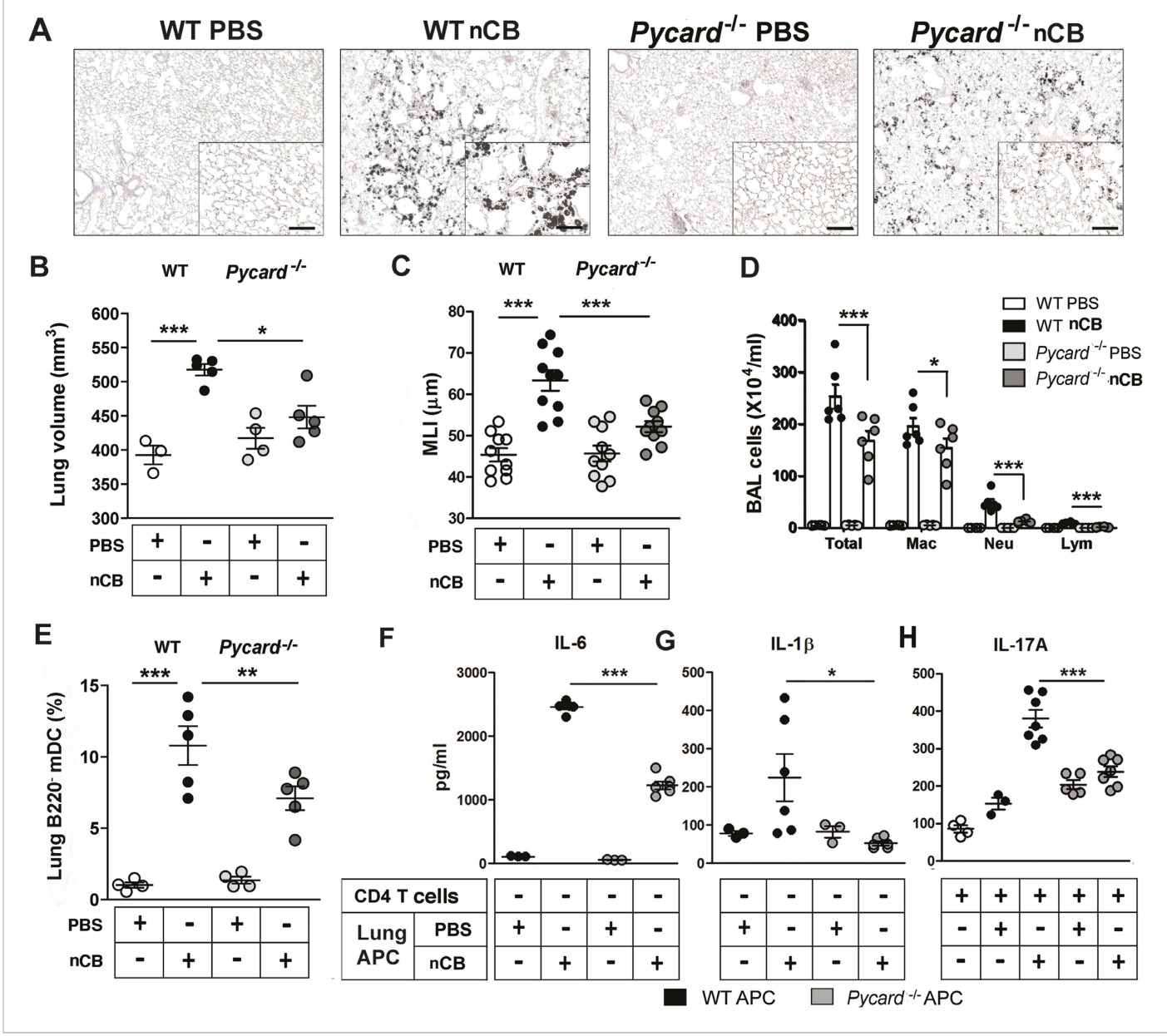

**Figure 6.** ASC-mediated inflammasome pathway is required for nCB-induced Th17 responses and emphysema. (**A**) Representative H&E staining of lung sections from WT and *Pycard*[−/−] mice exposed to nCB or vehicle (PBS) as described in *Figure 2—figure supplement 1*. Scale bar: 100 μm. (**B**) Micro-CT quantification of lung volume in indicated groups of mice. (**C**) Lung MLI measurement in the same group of mice. (**D**) Total and differential cell count in bronchoalveolar (BAL) fluid: macrophages (Mac), neutrophils (Neu), and lymphocytes (Lym). (**E**) Relative abundance of lung mDCs (CD11c[+]CD11b[high]) isolated from whole lung tissue in the same group of mice. IL-6 (**F**) and IL-1β (**G**) concentrations detected by ELISA in the supernatant of lung CD11c[+] cells isolated from indicated group of mice after overnight culture. (**H**) IL-17A concentration detected by ELISA in the supernatant of splenic CD4[+] T cells co-cultured with lung CD11c[+] cells isolated from indicated group of mice for 3 days in the presence of anti-CD3 (1 μg/ml). ***p < 0.001, **p < 0.01, *p < 0.05 as determined by the one-way ANOVA and Bonferroni's multiple comparison test; n = 3 to 7 per group, and data are mean ± SEM and representative of two independent studies.

The following figure supplement is available for figure 6:

**Figure supplement 1.** *Pycard*[−/−] mice produce less pro-inflammatory chemokines in the lungs in response to nCB challenge.

that is universally found in the lungs of smokers with emphysema in driving this pathological response. Although several chemical identities have been proposed, our study is the first to clearly identify the anthracotic pigment as nCB and show that it accumulates specifically in human lung phagocytic cells.

Our functional studies are also the first to show that nCB administered to the lungs in pathophysiologically relevant amounts can induce sterile inflammation and emphysema that is indistinguishable from disease induced by exposure to cigarette smoke in mice. Thus, nCB is likely the major component of smoke that causes long-term lung toxicity. Furthermore, our findings have major implications regarding the safety of activities involving the chronic inhalation of smoke and the need to control the particulate composition of air. Since nCB is used extensively in the rubber, plastics, and composites industries, the exposure levels should also be controlled in the workplace.

Our findings also elucidate both the nature of and the mechanism by which inflammation in the lungs of heavy cigarette smokers is perpetuated even long after cessation of cigarette smoking. Both chronic exposure to cigarette smoke and inhalation of nCB mediate similar inflammatory responses that are characterized by the activation of lung mDCs, differentiation and accumulation of Th17 cells, and lung parenchymal destruction (emphysema) (*Shan et al., 2014*). In part, this sterile inflammatory response to nCB is due to activation of the inflammasome pathway. Specifically, we found that inhaled nCB induces the production of IL-1β and IL-6, two pro-inflammatory cytokines that are required for mDC-mediated differentiation of Th17 cells and emphysema development. This inflammation persists and lung damage continues to accumulate even after smoking cessation due to the insoluble nature of nCB. Although readily taken up by phagocytic lung cells that could theoretically be expectorated or migrate out of the lungs via the lymphatics (*Corry et al., 1984*), these cells most likely undergo cell death too rapidly in response to nCB ingestion for any of these potential clearance mechanisms to operate efficiently. The nCB is then released in the lung by the cells it kills, only to be taken up again and kill subsequent phagocytes. nCB thus establishes an unending cycle of cell death that, if sufficiently pronounced, will trigger activation of the inflammasome pathway in response to the release of danger-associated molecular patterns (DAMPs) from dying cells (*Piccinini and Midwood, 2010*). Immune responses both rapidly kill invading pathogens and solubilize antigenic and adjuvant-like pathogen-derived substances to facilitate their removal and thus terminate the potentially deleterious inflammation. Both of these fundamental immune functions are thwarted in the context of nCB accumulation, leading to a perpetual cycle of lung inflammation and damage.

Our findings, therefore, raise concerns that other insoluble environmental nanoparticles may, if inhaled, accumulate in lung phagocytic cells and induce similar pathology. In support of this, inflammasome-activated IL-1β has been shown to play a major role in lung sterile inflammation induced by other nanoparticles associated with lung diseases (*Merget et al., 2002*). ASC is required for the assembly of pro-caspase 1 in order to yield caspase 1 for the activation of pro-IL-1β to IL-1β (*Franchi and Nunez, 2012*). We show that inhaled nCB can activate the inflammasome pathway that results in production of mature IL-1β. In addition, inflammasome sensors activated by nCB-damaged cells require ASC activation because lung mDCs isolated from *Pycard*$^{-/-}$ mice failed to increase IL-1β and showed attenuated Th17 responses. A critically important physical feature of nCB, accounting in large part for its pro-inflammatory potential, is its hydrophobic character. Exposure to large quantities of hydrophobic nCB has been shown to induce cell injury, pyroptosis and generate reactive oxygen species (ROS) in cultured cells (*Reisetter et al., 2011*). One particular characteristic of nCB that correlates with its toxicity is its large surface area; larger forms of elemental carbon have much less potential to induce cell injury (*Oberdörster et al., 2005*) and, as we have shown here, damage DNA.

A single burning cigarette can generate approximately $10^{12}$ particles that vary in size from 1 micron to a few nanometers in diameter (*Sahu et al., 2013*). The deposition site of particulate matter in the lungs of smokers is governed largely by size, with larger particles depositing in the mouth and upper airway while smaller particles are deposited in progressively smaller and more distal airways (*Adam et al., 2006*; *Baker and Dixon, 2006*). For our studies, we used nCB spheroids with a nominal size of 15 nm that aggregate in clusters of 3–4, forming 50–75 nm per particle. However, in aqueous solution, this material forms macro-aggregates that fail to distribute evenly in the lung after intranasal challenge as does nCB delivered by smoke inhalation. We were partially successful in alleviating this confounding factor by adding sucrose to the nCB in aqueous solution. Nonetheless, although we endeavored to deliver nCB to mice in amounts that matched actual burdens found in human lung, it is likely that we did not fully recapitulate the in vivo particle size and distribution of nCB acquired through smoke inhalation. Further studies are required to define how nCB size affects in vivo toxicity as defined in these studies.

Thermal and chemical analyses have shown that a combustion heat of 350–550℃ yields black carbon (BC) that contains PAHs that are linked to inflammation (*Bleck et al., 2006*), but combustion at

much higher temperatures (650–1100°C) produces PAH-free CB (*Watson et al., 2005*); we used this material heated to higher temperatures for intranasal administration in mice. A critical question related to our studies is, therefore, which form of carbon, BC or CB, is deposited in the lung during smoking, and how much do PAHs contribute to smoking-related lung inflammation. The nCB used in our studies lacked detectable PAHs by gas chromatography mass spectrometry analysis, which has a sensitivity limit of 1 part in $10^{10}$ by mass. While PAHs are present in soot and diesel exhaust particles (*Garza et al., 2008*), neither Raman spectroscopy nor hyperspectral imaging can distinguish CB from BC as could be found in human lungs. However, X-ray analysis of the melting of small metal particles has revealed the temperature distribution inside a burning cigarette as 850–920°C during active inhalation, decreasing to 700°C during the smoldering phase; this temperature regime is consistent with the creation of predominant CB during smoking (*Baker, 1974*). Moreover, as PAH-free CB recapitulates almost entirely the pathology induced by smoke exposure, we conclude that sterile inflammation and pulmonary emphysema are primarily the result of CB accumulation in the lung and not that of BC or PAHs therein during smoking.

Among signaling pathways that are important for APC activation and inflammation, we found that phosphorylation of Erk was significantly upregulated in cells exposed to CB treatment in a dose- and time-dependent manner while phosphorylation of p38 or JNK were not changed. The most notable, and unexpected, result of our RPPA analysis was the upregulation of several DNA damage enzymes by nCB exposure. This led us to confirm that nano-sized CB induces DSB in DNA as detected by the expression of γH2AX, an ATM-regulated pathway. We show that larger forms of CB result in attenuation of DSB, indicating that the size of CB is an important factor in its genotoxicity. Activation of this DNA repair pathway is in turn linked to the production of pro-inflammatory cytokines such as IL-6 and IL-1β that are required for Th17 differentiation. In addition, microRNA-22 (miR-22) has been shown to be upregulated in lung APC of mice exposed to smoke or nCB and is critical in Th17 responses through activation of AP-1 complexes and histone deacetylase (HDAC) 4 (*Lu et al., 2015*). Thus, together with DAMPs released by cells killed by nCB, nCB-induced DNA damage accounts for much of the inflammatory nature of nCB. The mechanism by which nCB cleaves DNA and the additional biological consequences of this adverse property remain active areas of investigation in our laboratory.

In summary, our findings show that inhalation of cigarette smoke leads to the accumulation of nCB in airway APCs (macrophages and mDCs). This insoluble material promotes perpetual Th17 cell-mediated lung inflammation in part through the double-stranded cleavage of nuclear DNA. These findings largely explain the persistent and incurable nature of smoking-related lung disease. Because no medical means of removing accumulated lung nCB exists, our findings underscore the need for all individuals and societies to minimize the production of and exposure to smoke-related particulate air pollution and industrial nCB.

## Materials and methods

### Mice

C57BL/6J mice were purchased from the Jackson Laboratory (Bar Harbor, ME). *Pycard*$^{-/-}$ mice (C57BL/6 background; Pycard encodes for Asc protein [*Mariathasan et al., 2004*]) were obtained from Dr Vishva Dixit (Genentech, South San Francisco, CA). Il-17a$^{-/-}$ mice (C57BL/6 background) were obtained from Dr Chen Dong (The University of Texas MD Anderson Cancer Center, Houston, TX). All mice were bred in the transgenic animal facility at Baylor College of Medicine. All experimental protocols (AN-4589) used in this study were approved by the Institutional Animal Care and Use Committee of Baylor College of Medicine and followed the National Research Council Guide for the Care and Use of Laboratory Animals.

### Reagents

MEK1/2 inhibitor U0126 was purchased from Cell Signaling (Danvers, MA). DNA-PKc inhibitor Nu7026 was purchased from Tocris (Bristol, UK). ATM inhibitor Ku55933 was purchased from Millipore (Billerica, MA). Pierce LDH Cytotoxicity Assay Kit was purchased from Life Technologies (Grand Island, NY), and LDH release was measured according to the manufacturer's instructions.

## Nanoparticle characterization and preparation

Various sizes of carbon black nanoparticles (nCB) were obtained from Cabot Corporation (15 nm, Monarch 1100, Lot 1278105; 35 nm, Vulcan 9A32, Lot: CS-5822; 70 nm diameter, Sterling NS1, Lot 761510, CAS# for CB: 1333-86-4; Alpharetta, GA). The nCB, although listed to be 15 nm, is more precisely described by the manufacturer to have 15-nm CB particles that are arranged in clusters of 3–5 particles, much as grape clusters, so the actual size is ~50–75 nm diameter clusters. Conjugation of polyethylene glycol to nCB (PEG-nCB) was performed as described (Zhou et al., 2014). Briefly, 15-nm nCB (250 mmol) was dispersed in tetrahydrofuran (THF) using bath sonication for 3 hr. Then, 4,4′-azobis(4-cyanopentanoic acid) (ACPA) was added to the nCB dispersion in a three-step process. The first portion of ACPA (7 mmol) was added and stirred continuously for 24 hr at 70°C. This addition was repeated at 24 hr and 48 hr. The mixture was cooled to room temperature, filtered through a 45-μm pore Teflon membrane, washed with THF, ethanol and acetone for three times, and vacuum dried (60°C, 100 Torr), producing carboxyl-functionalized nCB. Carboxyl-functionalized nCB (2 mmol) was dispersed in dimethylformamide for 30 min, then mixed with $N,N'$-dicyclohexylcarbodiimide (0.8 mmol), mPEG-NH$_2$ (0.04 mmol), and dimethylaminopyridine (2 flakes). The reaction was stirred for 24 hr, transferred to a dialysis bag (molecular weight cut-off, 5.0 kDa), dialyzed in running deionized water for 1 week, and filtered through a 0.22-μm pore Teflon membrane. X-ray photoelectron spectroscopy performed on an a PHI Quantera SXM scanning X-ray microprobe with 26 eV passing energy, 45° takeoff angle, and a 100-μm beam size. Thermogravimetric analysis performed on a TA Instruments Q-600 Simultaneous TGA/DSC. FTIR spectra were recorded using a Nicolet FTIR with an ATR attachment. By TGA, PEG-nCB contains 42–50% PEG by mass. By X-ray photoelectron spectroscopy, the PEG-nCB surface is 15% carbon, 56% oxidized carbon, 1.2% nitrogen, and 28% oxygen. FTIR: 840(m), 960(m), 1060(w), 1090(vs), 1150(m), 1240(m), 1280(m), 1340(m), 1360(m), 1470 (m), 1570(w), 1620(w), 2880(s) cm$^{-1}$.

## Animal model of nCB-induced emphysema

We suspended 20 mg of endotoxin-free nCB or 47 mg PEG-nCB (containing 20 mg of nCB by weight) in pre-warmed 1 ml *tert*-butyl alcohol with 1% sucrose. The mixture was then frozen at −80°C for 24 hr and was placed on a vacuum pump and lyophilized until dry for 24 hr. nCB or PEG-nCB mixed with 1% sucrose were rehydrated in sterile PBS to achieve the concentration of 0.5 mg/50 μl, vortexed and sonicated in a water bath for 10 min before administration. Mice were deeply anesthetized with isoflurane, and 50 μl droplets containing 1% sucrose in PBS (vehicle) or nCB or PEG-nCB (mixed with 1% sucrose in PBS) were applied to the nares. Mice were challenged twice per week for 6 weeks and were sacrificed 1 month after the last challenge.

## Quantification of experimental model of emphysema

The severity of lung parenchymal destruction (emphysema) was determined by computed tomography (CT) methods (Shan et al., 2012). Briefly, mice were anesthetized with etomidate (30 mg/kg) and placed in an animal CT scanner (Gamma Medica, Salem, NH), and completed images of the chest were obtained by the Animal Phenotyping Core in Baylor College of Medicine. Amira 3.1.1 software (FEI, Hillsboro, OR) was used to process the images and quantification of emphysema in three dimensions.

The MLI used for measurement of mouse lung destruction (e.g., lung morphometry) was calculated as previously described (Shan et al., 2014). Briefly, an unbiased observer randomly selected ten fields from the left lobe (large airways and vessels were excluded). Paralleled lines were placed on serial lung sections and MLI was calculated by multiplying the length and the number of lines per field, divided by the number of intercepts.

## Analysis of experimental model of emphysema

BALF and lung tissue were collected as previously described (Goswami et al., 2009). Briefly, mice were anesthetized with etomidate and BALF was collected by instilling and withdrawing 0.8 ml of sterile PBS twice through the trachea. Total and differential cell counts in the BALF were determined with the standard hemocytometer and HEMA3 staining (Biochemical Sciences Inc, Swedesboro, NJ) using 200 μl of BALF for cytospin slide preparation. Mouse lungs were dissected to prepare single-cell suspensions; alternatively, lungs were fixed with instillation of 4% paraformaldehyde solution via a

tracheal cannula at 25-cm $H_2O$ pressure followed by paraffin embedding and were sectioned for histopathological studies. Hematoxylin and eosin (H&E) staining was performed as described (*Goswami et al., 2009*).

## Intracellular cytokine staining

Mouse lung RBC-free single-cell suspension were stimulated with phorbol 12-myristate 13-acetate (PMA, 10 ng/ml; Sigma–Aldrich, St. Louis, MO) and ionomycin (1 µg/ml; Sigma–Aldrich) for overnight supplemented with brefeldin A (10 µg/ml; Sigma–Aldrich) for the last 6 hr. Cells were stained for surface markers with anti-CD3, anti-CD4, anti-CD8, and anti-γδTCR antibodies and then fixed with FACS lysing solution (BD BioSciences, San Jose, CA), permeabilized with 0.5% saponin (Sigma–Aldrich), and stained with anti-IFNγ and anti-IL-17A antibodies for analysis of intracellular cytokine production by flow cytometry.

## Mouse immune cell isolation from lung, spleen, and bone marrow-derived dendritic cell (BMDC) culture

Mouse lung or spleen single-cell suspensions were prepared by mincing whole organs through a 40-µm cell strainer (BD Falcon, San Jose, CA) followed by red blood cell (RBC) lysis (ACK lysis buffer, Sigma–Aldrich) for 3 min. For isolation of lung APCs, RBC-free whole lung cells were labeled with anti-CD11c-conjugated magnetic beads (Miltenyi Biotec, San Diego, CA) and then isolated by autoMACS (Miltenyi Biotec). For isolation of spleen CD4 T cells, RBC-free whole splenocytes were labeled with anti-CD4 conjugated magnetic beads (Miltenyi Biotec) and then isolated by autoMACS. Mouse BMDCs were prepared as previously described with some modification (*Lutz et al., 1999*) Femurs and tibias of 4- to 8-week-old female were isolated and freed from the surrounding tissue. Intact bones were kept in 70% ethanol for 3 min followed by a PBS wash. Both ends of the bones were cut with scissors, and the marrow was flushed out with RPMI-1640 medium through a syringe with 26.5 needle. RBCs were then removed by ACK lysis buffer and cell debris or tissue clusters were filtered out. Cells from bone marrow were cultured in a 6-well plate with 20 ng/ml mouse GM-CSF and 10 ng/ml mouse IL-4 (R&D Systems, Minneapolis, MN) for 5 to 6 days.

## Human immune cell isolation from lung and human MDDC culture

Human lung single cell suspensions were prepared as previously described (*Shan et al., 2009*). Briefly, fresh lung tissue was cut into 0.1-cm pieces in Petri dishes and treated with 2 mg/ml of collagenase D (Roche Pharmaceuticals, Basel, Switzerland) in HBSS and incubated for 30 to 40 min at 37°C. Single cells were collected by mincing the digested lung tissue through a 40-µm cell strainer (BD Falcon) followed by RBC lysis. Lung CD1a$^+$ DCs were isolated by labeling RBC-free lung cells with anti-CD1a-conjugated magnetic beads (Miltenyi Biotec) and then isolated by autoMACS. PBMCs were isolated by Ficoll–Paque (GE Healthcare Life Sciences, Pittsburgh, PA) density gradient centrifugation. Human MDDCs were prepared as previously described (*Shan et al., 2009*). Briefly, RBC-free PBMCs were seeded in 6-well plates for 2 hr at 37°C and then nonadherent cells were removed by washing with PBS. Adherent cells were cultured with 50 ng/ml human GM-CSF and 10 ng/ml human IL-4 for 5 to 6 days.

## In vitro nanoparticle treatment, APC and T cell co-culture and cytokine measurement

CD11c$^+$ cells isolated from mouse lung, BMDCs, monocyte-derived (MD)DCs or RAW 264.7 cells (mouse leukemic monocyte/macrophage cell line) (ATCC, Manassas, VA) were treated with indicated amount of nCB for 1 or 2 days, were washed and placed in co-culture assays with or without T cells (at 1:10 ratio). Mouse APCs were co-cultured with congenic splenic CD4$^+$ T cells (1:10 ratio) in the presence of anti-mouse CD3 (1 µg/ml; BD Biosciences) for 3 days. ELISA (BD BioSciences) or Multiplex kit (Millipore) were used for the measurement of concentration of IL-17A, IFNγ, IL-4, IL-6, IL-1β, IL-1α, IL-12p70, TNFα, MIP-1α, MIP-1β, KC, RANTES, MCP-1, IP-10 in either lung homogenate or supernatant collected from cultured cells.

## siRNA transfection

Mouse ATM siRNA and scramble siRNA were purchased from Sigma–Aldrich. Mouse RAW 264.7 cells were transfected with Nucleofection kit (Lonza, Basel, Switzerland) according to the manufacturer's instructions. RAW cells treated with siRNA were incubated for 6 hr before CB treatment for overnight.

## Flow cytometry and antibodies

Flow cytometry was performed with a BD LSR II (BD BioSciences), and data were analyzed with FlowJo software (Tree Star Inc., Ashland, OR). The following anti-mouse antibodies were purchased from BD Pharmingen and used: Pacific Blue-CD3 (500A2), PE-Cy5-CD4 (RM4-5), APC-Cy7-CD8 (53-6.7), PE-IL-17A (TC11-18H10) and APC-IFNγ (XMG1.2). FITC-γδTCR (eBioGL3), eFluro450-B220 (RA3-6B2), PE-CD11b (M1/70), and APC-CD11c (N418) were purchased from eBioscience (San Diego, CA) and used.

## Western blot

RAW 264.7 cells or BMDCs were harvested, pelleted, washed with PBS and lysed in RIPA (Radio-immunoprecipitaiton Assay) buffer (Sigma–Aldrich) with a cocktail of proteinase and phosphatase inhibitor (Thermo Scientific, Waltham, MA). The protein concentration of whole cell lysate was detected by BCA kit (Thermo Scientific). Equivalent amounts of protein in each sample were resolved by SDS-PAGE and transferred into nitrocellulose membranes. Membranes were blocked in 5% nonfat-dried milk in PBS with 0.05% Tween 20. Rabbit anti-mouse phospho-Erk (Cell Signaling, Danvers, MA) was used for protein detection.

## Immunostaining

Cytospins of single cell suspensions were fixed with 4% formaldehyde, permeabilized with 0.5% saponin, and blocked with 3% BSA and Fc receptor Blocker (BD BioSciences). Then cells were stained with anti-γH2AX (Millipore) for overnight and detected by antibodies labeled with DAPI (4′,6-diamidino-2-phenylindole) and Alexa Fluor 488. Images were detected with Nikon ECLIPSE TE2000 and NIS-Elements software version 2.30 and Leica DFC300 FX.

## RPPA analysis of RAW 264.7 cells treated with nanoparticles

RPPA analysis was performed at the University of Texas MD Anderson Proteomic Core facility. Control RAW cells (untreated) and nanoparticle treated (100 µg/ml nCB and 100 µg/ml PEG-nCB) for 24 hr in triplicates were washed, pelleted, and subjected to RPPA analysis. A detailed description of sample processing and data analysis is available on the website of the core facility. Heatmaps were generated by the softwares Cluster and Treeview.

## In vitro mouse CD4 T cell differentiation

Naive CD4$^+$ T cells were isolated using anti-CD4-conjugated magnetic beads (Miltenyi Biotec) and were isolated with an autoMACS cell separator. Cells were differentiated under Th1, Th17, or Treg polarizing conditions. In brief, 2 to $2.5 \times 10^6$/ml cells were activated with 1.5 µg/ml plate-bound anti-CD3 and 1.5 µg/ml soluble anti-CD28 antibodies in addition to: 10 µg/ml anti-IL-4 antibodies, 50 U/ml IL-2 and 20 ng/ml IL-12 (Th1 polarizing condition), or 10 µg/ml anti-IL-4 antibodies, 10 µg/ml anti-IFNγ antibodies, 50 U/ml IL-2, 40 ng/ml IL-6 and 6 ng/ml TGFβ (Th17 polarizing conditions) or 10 µg/ml anti-IL-4 antibodies, 10 µg/ml anti-IFNγ antibodies, 50 U/ml IL-2 and 6 ng/ml TGFβ (Treg polarizing conditions). In some experimental groups 100 ng/ml nCB or vehicle control were added. Cells were cultured for 3 to 5 days, were harvested and washed for intracellular staining of IL-17, IFNγ or Foxp3 and the surface staining of CD25 to determine Th17, Th1, or Treg differentiation.

## mRNA isolation and quantitative PCR

Cell pellets were treated with TRIzol (Life Technologies), and mRNA was extracted as previously described (*Shan et al., 2014*). All probes, mouse Mmp9 (Mm00600164_g1), mouse Mmp12 (Mm00500554_m1), mouse Il6 (Mm00446190_m1), mouse Atm (Mm01177457_m1), and mouse Tnf (Mm00443258) were purchased from Applied Biosystems (Foster City, CA). All data were normalized to 18S ribosomal RNA (Hs99999901_s1) expression.

## TEM and high-resolution TEM

TEM: human lung CD1a[+] cells were fixed with modified Karnovsky's fixative in 0.1 M Millonig's phosphate buffer, osmicated, minced into 1-mm cubes, dehydrated, and stained with saturated uranyl acetate (*Watson, 1958*). Cells cubes were embedded in resin and polymerized at 68°C for 2 days (*Spurr, 1969*). Sections of 55–65 nm were cut and collected on 150 hex-mesh copper grids and counterstained with Reynold's lead citrate (*Reynolds, 1963*). The stained sections were viewed with a Hitachi H7500 transmission electron microscope, and images were captured by Gatan US1000 digital camera and Digital Micrograph, v1.82.366 software.

HR-TEM: the human emphysematous lung tissue was digested with proteinase K overnight followed by wash with ethanol and air-dried. The residual black substance was drop-cast directly on a lacey carbon TEM grid (Ted Pella, Inc., Redding, CA) and vacuum dried for 6 hr before usage. The HR-TEM image was taken with the JEM-2100F field emission gun transmission electron microscope (JEOL USA, Inc., Peabody, MA) operated at 200 kV.

## Hyperspectral mapping of dark field imaging

Samples were imaged and analyzed by CytoViva. Darkfield hyperspectral imaging was performed using a CytoViva dark field microscope system equipped with CytoViva Hyperspectral Imaging System 1.2 (Auburn, AL).

## Raman imaging

The Raman spectrum of 15 nm carbon black from Cabot (Lot: 1278105) was acquired using a Renishaw inVia Raman Microscope (Hoffman Estates, Il) with a 514.5-nm laser and a 50× objective lens. Similarly, Raman spectra were acquired from individually dispersed cells drop cast onto a glass slide and was fixed with 4% paraformaldehyde. The spectra from each cell type were combined, and background cellular fluorescence was subtracted. Raman maps were taken with a pixel size of 2 μm × 2 μm.

## GCMS of PAHs in CB

Soxhlet extraction with dichloromethane and direct extraction with ODCB were used to determine PAH contamination of CB (*Harwood and Moody, 1989*). GCMS was used to probe for PAHs, which failed to detect below the limit of detection (1 part in $10^{10}$ by mass) as standardized by Pyrene and Anthracene (*Pilla et al., 2009*).

## Statistical analysis

For the comparison of cytokine production and gene expression of mice challenged with nanoparticles, cells treated with nanoparticles and reagents, and CT quantification of mouse lung volume, we used the Student's *t*-test or one-way analysis of variance (ANOVA) test and Bonferroni's multiple comparison test. All data shown are the mean ± standard error of the mean (SEM), and all analyses were performed with the Prism software (GraphPad Software).

## Acknowledgements

We thank Alexander Seryshev and Joel M. Sederstrom for his technical assistance. This project was supported by the Cytometry and Cell Sorting Core at Baylor College of Medicine, with funding from the NIH (NIAID P30AI036211, NCI P30CA125123, and NCRR S10RR024574), and the Integrated Microscopy Core at Baylor College of Medicine for TEM image acquisition and microscope assistance, supported by grants 1081701321-P30-CA, 1081701233-DLDCC, 1081701347-U54-HD, 1081701347-P30-DK, and the Digestive Disease Center grant.

## Additional information

### Funding

| Funder | Grant reference | Author |
| --- | --- | --- |
| National Institutes of Health (NIH) | HL117181 | David B Corry, Farrah Kheradmand |

| Funder | Grant reference | Author |
|--------|-----------------|--------|
| VA Merit Award | | David B Corry, Farrah Kheradmand |

The funders had no role in study design, data collection and interpretation, or the decision to submit the work for publication.

## Author contributions

RY, WL, MS, ELGS, DCM, WKAS, XY, LS, AYH, FK, Conception and design, Acquisition of data, Analysis and interpretation of data, Drafting or revising the article; JMB, ZS, Conception and design, Acquisition of data, Analysis and interpretation of data, Contributed unpublished essential data or reagents; JMT, DBC, Conception and design, Analysis and interpretation of data, Drafting or revising the article, Contributed unpublished essential data or reagents

## Ethics

Animal experimentation: C57BL/6J mice were purchased from the Jackson Laboratory. ASC−/− mice (C57BL/6 background) were obtained from Dr Vishva Dixit (Genentech, South San Francisco, CA). IL-17A−/− mice (C57BL/6 background) were obtained from Dr Chen Dong (The University of Texas MD Anderson Cancer Center, Houston, TX). All mice were bred in the transgenic animal facility at Baylor College of Medicine. All experimental protocols (AN-4589) used in this study were approved by the Institutional Animal Care and Use Committee of Baylor College of Medicine and followed the National Research Council Guide for the Care and Use of Laboratory Animals.

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
