## [Decision Letter]

Thank you for submitting your work entitled “Nanoparticulate Carbon Black in Cigarette Smoke Induce DNA Cleavage and Th17-Mediated Emphysema” for peer review at *eLife*. Your submission has been favorably evaluated by Charles Sawyers (Senior Editor) and three reviewers, one of whom is a member of our Board of Reviewing Editors. Two of the three reviewers, Edwin Blalock and Dean Sheppard, have agreed to share their identity.

The reviewers have discussed the reviews with one another and the Reviewing editor has drafted this decision to help you prepare a revised submission.

This paper describes a remarkable series of studies that fundamentally change our understanding of the toxicity of nanoparticulate carbon black (nCB) and its role in sterile inflammation and emphysema. The authors show that nCB accumulates in myeloid dendritic cells in lungs from emphysema patients and also in antigen presenting cells (APC) in a smoking mouse model of emphysema. Like cigarette smoke, nCB administration into the always induced emphysema in mice and such nCB persisted indefinitely in mouse lungs. nCB apparently causes disease by activating lung APC which promote Th17 helper cell differentiation via double stranded DNA break and ASC-mediated inflammasome assembly in phagocytes. The adverse effects of nCB can be negated by increasing particulate size or decreasing hydrophobicity.

Overall, this paper epitomizes translational medicine. It covers the gamut from state-of-the-art chemistry to demonstrate that nCB is a prominent component of smoke toxicity and is the likely anthracotic pigment in emphysematous lungs to showing that nCB can cause emphysema in mice. Furthermore, the authors work out the molecular as well as cellular mechanisms of nCB's toxic effects. Lastly, the authors suggest ways by which nCB toxicity can be mitigated. Experimentally the study is a tour-de-force of state-of-the-art technologies that have yielded very convincing results. These in turn have provided a paradigm shift in our understanding of a key environmental pollutant.

However, there are a few pending issues that we would like to see addressed in a revised submission:

1) As for the method used to assess the presence of emphysema, most papers in this field utilize more than one method to increase the confidence that emphysema is present and accurately measured. Without performing any new exposures, the authors could add data on mean linear intercept or another quantitative assessment of lung morphology and of lung density obtained from the same CT scans used to calculate lung volume. This is especially important given the relatively small size of the changes in lung volume seen and the small sample sizes. Moreover, the lung volume data should be shown without zero suppression on the y axis so readers could accurately see the small effect size.

2) It is nice that the authors showed much of their data as scatter plots rather than simply bar graphs, since these are much more informative representations of small samples. The authors should use this format of data presentation throughout the manuscript. It would also be preferable to present the data on T regs in Figure 3–figure supplements 3-6 as actual percentages rather than fold change, to give readers a better sense of the variability among control samples.

3) Support for the authors' conclusion that the pathway by which nCB promotes Th17 cell differentiation and emphysema involves DNA damage and Erk activation could be strengthened if the authors included evidence that inhibition of Erk signaling inhibits both IL-17A production and IL-6 production. It would also be helpful to provide evidence about whether or not these events were ordered in the same pathway, for example by determining whether inhibition of DNA damage responses decreases activation of Erk. It would also be helpful to assess the specificity of the pharmacologic effects shown, for example by providing parallel data on the effects of the same range of U0126 concentrations on Erk phosphorylation in the cells under study.

4) NLRP3 is the only inflammasome that is known to respond to particulate matter and danger signals in sterile inflammation. Therefore, the authors should perform the same experiment in *Nlrp3* knockout mice (should be easily accessible) similarly as they have done with the ASC knockout mice. This will make the inflammasome part more complete and convincing.

---

## [Author Response]

1) As for the method used to assess the presence of emphysema, most papers in this field utilize more than one method to increase the confidence that emphysema is present and accurately measured. Without performing any new exposures, the authors could add data on mean linear intercept or another quantitative assessment of lung morphology and of lung density obtained from the same CT scans used to calculate lung volume. This is especially important given the relatively small size of the changes in lung volume seen and the small sample sizes. Moreover, the lung volume data should be shown without zero suppression on the y axis so readers could accurately see the small effect size.

We appreciate the referees’ suggestion to include a second method to assess emphysema. The quantitative CT lung volume measurements in this protocol have been validated independently using mean linear intercept (MLI) data ([45], [46], Yuan et al. 2015). However in response to this comment, we have revised Figures 2, 3, 4 and 6 that now include corresponding MLI measurements and confirm lung destruction in addition to the quantitative CT data.

In response to the second concern regarding the effect size of nCB on lung parenchyma destruction (emphysema), we show here that mice exposed to nCB develop approximately 25 to 35% emphysema when compared to PBS control. This change in lung volume (% emphysema) is statistically significant, and is slightly greater than exposure to 4 months of cigarette smoke in the same strain of mice (45). These findings couldn’t be directly compared to smoke-induced lung disease in humans, nevertheless lung destruction at 20-30% is strongly associated with end-stage emphysema in smokers (Coxson et al. 2013, Hesselbacher et al. 2011). Further, in this report, as well as in prior publications, we show lung volume data in such a way to minimize the blank space in the figures (Shan 2014 et al., Yuan et al. 2015). Changing this parameter will result in inclusion of a larger area of space in each figure (Figure 7). Therefore, to keep the figure flow consistent, we used the zero suppression; however if our reviewers feel that leaving blank spaces in the figures is necessary, we would revise all 8 panels accordingly.

Author response image 1.Lung volume and MLI figures without zero suppression.**DOI:**
http://dx.doi.org/10.7554/eLife.09623.029

2) It is nice that the authors showed much of their data as scatter plots rather than simply bar graphs, since these are much more informative representations of small samples. The authors should use this format of data presentation throughout the manuscript. It would also be preferable to present the data on T regs in Figure 3–figure supplements 3-6 as actual percentages rather than fold change, to give readers a better sense of the variability among control samples.

We have revised our figures to show individual data points throughout the manuscript. In response to the reviewers’ concern regarding small sample size, we would like to point out that while each panel in the figures may show data points for 4 to 6 mice in each group, the data are representative of at least 2 or 3 independent experiments. Therefore our findings are statistically significant within each experiment and are reproducible. We have revised the Figure 3—figure supplement 3, Figure 3—figure supplement 4 and Figure 3—figure supplement 5 to show percentages of Tregs from four independent experiments. We hope that these changes address the reviewers’ concerns.

3) Support for the authors' conclusion that the pathway by which nCB promotes Th17 cell differentiation and emphysema involves DNA damage and Erk activation could be strengthened if the authors included evidence that inhibition of Erk signaling inhibits both IL-17A production and IL-6 production. It would also be helpful to provide evidence about whether or not these events were ordered in the same pathway, for example by determining whether inhibition of DNA damage responses decreases activation of Erk. It would also be helpful to assess the specificity of the pharmacologic effects shown, for example by providing parallel data on the effects of the same range of U0126 concentrations on Erk phosphorylation in the cells under study.

In order to address this question, lung CD11c^+^ APCs from PBS or nCB-challenged mice were co-cultured with naïve splenic CD4 T cells in the presence or absence of increasing concentrations of U0126 (MEK inhibitor). We found that IL-17A, and IL-6 concentrations were significantly reduced in APCs of nCB-challenged mice in a dose dependent manner (Figure 8). These findings support the hypothesis that Erk phosphorylation is required for both IL-17A and IL-6 production.

Author response image 2.Inhibition of Erk phosphorylation reduces IL-17A and IL-6.CD11c^+^ lung cells were isolated from lungs of mice exposed to PBS or nCB as described in the manuscript and treated with increasing concentration of U0126 for two days (left panel). Using the same conditions cells were co-cultured with congenic splenic T cells (1:10 ratio) and antiCD3 (1μg/ml) in the presence of increasing concentration of U0126 for three days (right panel). Concentration of IL-17A, and IL-6 were measured using ELISA. ** p<0.01, *** p<0.001 as determined by the one-way ANOVA and Bonferroni’s Multiple Comparison test.**DOI:**
http://dx.doi.org/10.7554/eLife.09623.030

In response to the concern regarding the inhibition of DNA damage and specificity of Erk inhibition, we stimulated BMDCs with nCB (PBS was used as control) in the presence or absence of Ku55933 (ATM inhibitor), Nu7026 (DNA-Pkc inhibitor) or different concentration of U0126 (MEK inhibitor) for 24 hours. Western blot analysis of cell lysates showed decreased phosphorylation of Erk in response to Nu7026 and U0126 (in a dose-dependent manner) but not Ku55933. Therefore, inhibition of DNA damage can partially decrease activation of Erk (Figure 9).

Author response image 3.Nu7026 and U0126 inhibit Erk phosphorylation.BMDCs were treated with PBS or nCB in the presence of different inhibitors as indicated (Ku: Ku55933, 3μM; Nu: Nu7026, 3μM; U0: U0126, 0.1μM) for 24 hours and phosphorylated Erk (p-Erk), total Erk and β-actin were measured using western blot of cell lysates (left panel). Inhibition of p-Erk was examined using the same system in the presence of increasing concentration of U0126 (right panel).**DOI:**
http://dx.doi.org/10.7554/eLife.09623.031

*4) NLRP3 is the only inflammasome that is known to respond to particulate matter and danger signals in sterile inflammation. Therefore, the authors should perform the same experiment in* Nlrp3 *knockout mice (should be easily accessible) similarly as they have done with the* ASC *knockout mice. This will make the inflammasome part more complete and convincing.*

We appreciate the referees’ suggestion to confirm our findings using *Nlrp3* knockout mice. Interestingly, although NLRP3 is the most studied inflammasome in relation to danger signals, other molecules (e.g. AIM2), have been shown to recognize cytosolic dsDNA (Cavlar, Ablasser and Hornung 2012, Hornung et al. 2009, Hornung and Latz 2010). Because ASC is critical in the inflammasome-mediated IL-1β activation pathway, we opted to examine the general role of inflammasome using *ASC*^*-/-*^ mice and these mice became available to us through the generosity of Dr. Vishva Dixit (Genentech). Furthermore, at the time of our initial investigation *Nlrp3*^*-/-*^ mice were not commercially available. These mice have recently become available for purchase from the Jackson laboratory ($205.9/mouse priced in 8-2015), but we were informed that an order of 10 mice could only be shipped within approximately 2-3 months after the purchase. Because our nCB experimental protocol requires an additional of 2-3 months to complete and be analyzed (i.e., a total of six months delay) we believe that the excessive time required to fully address a confirmatory role for NLRP3 would be beyond the 2-month timeline provided to us for revision of the current manuscript. Therefore, while we agree with the need for future studies, we would like to ask the referees to allow this new work to be deferred to future studies.